# Neosaxitoxin Inhibits the Expression of Inflammation Markers of the M1 Phenotype in Macrophages

**DOI:** 10.3390/md18060283

**Published:** 2020-05-27

**Authors:** M. Cecilia Montero, Miguel del Campo, M. Bono, M. Valeska Simon, Julia Guerrero, Néstor Lagos

**Affiliations:** 1Membrane Biochemistry Laboratory, Faculty of Medicine, University of Chile, Santiago 8380453, Chile; ceciliamontero.vet@gmail.com (M.C.M.); miguel.delcampo@fucited.cl (M.d.C.); 2Laboratory of Immunology, Faculty of Sciences, University of Chile, Santiago 7810000, Chile; mrbono@uchile.cl (M.B.); valeskasimon@yahoo.es (M.V.S.); 3Program of Physiology, Faculty of Medicine and Department of Internal Medicine Hospital Clínico, University of Chile, Santiago 8380453, Chile; jguerrero@uchile.cl

**Keywords:** neosaxitoxin, inflammation, Nav channels, macrophages

## Abstract

(1) Background: Neosaxitoxin (NeoSTX) has been used as a local anesthetic, but its anti-inflammatory effects have not been well defined. In the present study, we investigate the effects of NeoSTX on lipopolysaccharide (LPS)-activated macrophages. (2) Methods: Raw 264.7 and equine PBMC cells were incubated with or without 100 ng/mL LPS in the presence or absence of NeoSTX (1µM). The expression of inflammatory mediators was assessed: nitric oxide (NO) content using the Griess assay, TNF-α content using the ELISA assay, and mRNA of inducible nitric oxide synthase (iNOS), interleukin-1β (IL-1β), and tumor necrosis factor-α (TNF-α) using a real-time polymerase chain reaction. (3) Results: NeoSTX (1 μM) significantly inhibited the release of NO, TNF-α, and expression of iNOS, IL-1β, and TNF-α in LPS-activated macrophages of both species studied. Furthermore, our study shows that the LPS-induced release of inflammatory mediators was suppressed by NeoSTX. Additionally, NeoSTX deactivated polarized macrophages to M1 by LPS without compromising its polarization towards M2. (4) Conclusions: NeoSTX inhibits LPS-induced release of inflammatory mediators from macrophages, and these effects may be mediated by the blockade of voltage-gated sodium channels (VGSC).

## 1. Introduction

Inflammation is the primary response of innate immunity to an infection or damage that begins with the secretion of a variety of cytokines, chemokines, lipid mediators and bioactive amines that trigger increased blood flow and increased capillary permeability that allow the recruitment of blood monocytes to damaged tissue where they will differentiate into macrophages [1,2] whose functions include tissue maintenance, regulation of inflammatory homeostasis, removal of senescent cells, tissue remodeling, repair, genesis and resolution of inflammatory processes [3].

The final stage of monocyte-macrophage differentiation is given by tissue conditions so that the same monocyte can be differentiated into different types of macrophages affected by the tissue environment [1,4]. This plasticity allows macrophages to polarize in M1 or “classic” macrophages that destroy various types of pathogens and in M2 or “alternative” macrophages that promote tissue proliferation and repair [5,6].

Exposure to damage-associated molecular patterns (DAMPs) and pathogen-associated molecular patterns (PAMPs) [2,5,7] activate PAMP recognition receptors (PRRs) present in innate immune cells such as macrophages. Among the latter, toll like receptors (TLR) 2 and 4 favor macrophage polarization towards the M1 phenotype [5]. This process begins when TLR4 recognizes LPS by triggering the activation of the canonical pathway of nuclear transcription factor kB (NF-kB), considered the most important regulator of pro-inflammatory gene expression [8], mediating the expression of inflammatory cytokines such as IL-1β, TNF-α, and iNOS [7,8,9,10].

On the other hand, IL-4, IL-10, and IL-13 polarize macrophages towards M2 through IL-4R type I and II receptors that mediate cell signaling pathways that involve STAT6, p38 MAPK, and ERK [11,12]. These pathways allow, among other things, the expression of arginase 1 (Arg1) that metabolizes arginine into ornithine (precursor to polyamides and collagen), promoting tissue repair and growth [5,10], transcription of IL-10, a potent anti-inflammatory cytokine [13,14], and sequestration of co-activating molecules of NF-kB [6,9,12].

Under physiological conditions, the inflammatory process involves the activation of pro-inflammatory and anti-inflammatory response that includes the production of pro-inflammatory mediators such as TNF-α together with the increase of anti-inflammatory mediators such as IL-10. This regulation is essential for excessive damage, host prevention, and inflammatory homeostasis recovery [2,8,15,16].

The loss of balance in favor of the M1 phenotype can trigger pathologies that involve musculoskeletal and immunological disorders that produce chronic pain [17,18]. Among the treatments used for chronic pain management are anesthetic drugs that block voltage-gated sodium channels (VGSC) [19,20,21] responsible for the onset and spread of action potential in nerve, muscle, and neuroendocrine cells.

Nine isoforms of VGSC named Nav 1.1 through Nav 1.9 have been identified [20,22,23]. From a structural point of view, each Nav has an α subunit organized in 4 homologous domains (I–IV), each of which contains 6 transmembrane α-helices (S1–S6). The loop between segments S5 and S6 constitutes the pore of the channel, and segment S4 of each domain acts as a load sensor [22,23].

While it is true that sodium channels are primarily associated with the support of electrogenesis in excitable cells, various investigations have confirmed the presence of Nav functions in cell types considered electrically non-excitable, such as astrocytes, microglia, peritoneal macrophages, and cancer cells [20]. Its functionality has been related to processes of phagocytosis, motility, the release of bioactive molecules, regulation of Na/K-ATPase activity, regulation of membrane potential (Vm), and ion channels, transport channels, and metastatic activity [20,24,25,26,27].

All the pharmacological agents that act on Nav have their receptors in the α subunit. There are at least 6 different receptor sites for neurotoxins and a specific receptor site for local anesthetics that block the canal. Our interest in this study focuses on a neurotoxin, non-peptide analog saxitoxin that binds to the receptor 1 site blocking the pore. This receptor site is formed by amino acid residues in the pore loop and is located directly on the extracellular side of the pore [20,22].

In the clinical field, recent publications from our laboratory (Laboratory of Membrane Biochemistry, University of Chile) have determined that the local infiltration of gonyautoxin and neosaxitoxin (NeoSTX), both paralytic phycotoxins derived from saxitoxin [20,21,28], are safe and effective for pain control in human pathologies [29,30]. These similar results were observed by Lobo et al. in 2015 [31] in a clinical safety study for NeoSTX. An example of this has been its use for the treatment of acute and chronic anal fissure [32], anal sphincter relaxation [33], chronic tension-type headache [34], bladder pain syndrome [35], management of pain in knee arthroplasty [36], as an enhancer of local anesthetics such as bupivacaine [37], as well as in the treatment of pain in the shin of thoroughbred racehorses caused by dorsal metacarpal disease, in which a significant risk reduction of the clinical signs of pathology with functional recovery of the limb is observed [38]. This anesthetic response of the paralytic phycotoxin NeoSTX could be accompanied by the restoration of inflammatory homeostasis. However, to date, no studies have been conducted in humans or animals that assess the mechanisms involved in the anti-inflammatory action of NeoSTX.

The objective of this study is to evaluate the effects of NeoSTX on inflammatory cytokines derived from macrophages of the M1 phenotype expression.

## 2. Results

### 2.1. NeoSTX Does not Modify Cell Viability in Macrophages

To determine if NeoSTX has an effect on the viability of cells exposed to it, 2 × 10^4^ RAW 264.7 cells were treated with NeoSTX at concentrations of 0.1, 1, 10, and 100 nM, and 1 and 10 µM for 24, 48, and 72 h. The AlamarBlue technique was used to assess cell viability [39]. Under experimental conditions, cell viability was greater than 90% regardless of the concentration of NeoSTX used and the exposure time (Figure 1).

### 2.2. Nav Characterization in Macrophages

#### 2.2.1. Equine PBMC Cells Express the Nav 1.5 and Nav 1.6 Isoforms

We evaluated the presence of Nav in a primary culture of equine cells derived from PBMC. First, we generated a series of specific primers for the Nav isoforms (1.1 to 1.9) described for the equine species (Appendix A). For the positive control, we used different equine tissues where different Nav have previously been described: heart, brain, dorsal root ganglion (DRG), and striated muscle. For the negative control, a liver sample was used. Subsequently, the expression of the Nav isoforms mRNA (1.1 to 1.9) in equine macrophages was quantified using RT–PCR, and the product obtained was amplified in agarose gel. 

As shown in Figure 2A, Nav 1.5 and Nav 1.6 isoforms were observed in equine PBMC. Interestingly, Nav 1.1, 1.2, 1.3, 1.4, 1.8, and 1.9 isoforms were not detected in our cells, as had been previously described by Carrithers et al. [5] (Appendix A).

Then, to assess Nav 1.6 protein expression, equine PBMC were treated with rabbit anti-mouse Nav 1.6 antibody and analyzed by flow cytometry. A fluorescence index of 1.4 was observed, corroborating the Nav 1.6 protein isoform expression in this species (Figure 2B). In addition, confocal microscopy analysis using the same antibody shows a signal, although weak, in the cellular cytoplasm.

#### 2.2.2. RAW 264.7 Cells Express the Isoform of the Nav 1.6 Channel

The presence of the Nav 1.6 channel in RAW 264.7 cells was determined by flow cytometry (CF). As shown in Figure 2C, the fluorescence index (IF) was 1.7 (mean of the sample/mean of the control isotype).

To determine the cellular distribution of the Nav 1.6 channels, confocal microscopy analysis was done. Figure 2D shows RAW 264.7 cells Nav 1.6 channels are widely distributed in the intracellular space.

#### 2.2.3. LPS Induces the Expression of Nav 1.6 but not of Nav 1.5

The exposure of RAW 264.7 cells and equine primary culture to LPS (100 ng/mL for 18 h) increased the Nav 1.6 protein expression (IF = 1.6 and 1.7, respectively). This was confirmed by measuring the Nav 1.6 mRNA expression in the equine primary culture cell (*p* = 0.004) (Figure 2E). However, no significant changes were observed in the expression of the messenger of Nav 1.5. 

### 2.3. Macrophages Incorporate NeoSTX

Since Nav 1.6 is expressed intracellularly, we proposed to assess whether NeoSTX is incorporated into the cell. For this, we analyzed RAW 264.7 cells cultured for 24 h with NeoSTX and subsequently permeabilized and exposed to a polyclonal rabbit anti-NeoSTX antibody (Science and Technology Foundation for Development, FUCITED, Chile) and to the isotype anti-rabbit antibody control (FITC). Non-permeabilized cells were used as the control group. From the analysis by flow cytometry, we observed that the non-permeabilized cells did not show the presence of fluorescent marks in the cell membrane (IF 0.9) and that in the permeabilized cells, an IF of 25.5 was observed (Figure 3).

### 2.4. NeoSTX Inhibits Polarization Towards the M1 Macrophage Phenotype

In order to validate the results obtained in this objective, the effect of NeoSTX was compared with the effect of lidocaine, a specific voltage-dependent sodium channel blocker and known inhibitor of the expression of macrophage inflammatory cytokines exposed to an M1 inducer [40,41].

#### 2.4.1. Effect of NeoSTX and Lidocaine on RAW 264.7 Cells

In order to assess whether NeoSTX and lidocaine could induce mRNA expression of markers of inflammation, cells were cultured in the presence of NeoSTX (1 µM) or lidocaine (20 µg/mL) [40] for 24 h. As a result, it was observed that neither NeoSTX nor lidocaine induces the expression of TNF-α mRNA with respect to the control (*p* = 0.3094 and 0.3316, respectively), nor the production of NO (*p* = 0.1911 and 0.1842, respectively) nor of TNF-α (*p* = 0.1745 and 0.3198, respectively) in the culture supernatant with respect to the control (Appendix A). 

#### 2.4.2. Effect of NeoSTX and Lidocaine in the Primary Culture

In order to assess whether NeoSTX and lidocaine could induce mRNA expression of markers of inflammation, cells were cultured in the presence of NeoSTX (1 µM) or lidocaine (20 µg/mL) for 24 h. As a result, it was observed that neither NeoSTX nor lidocaine induces the expression of TNF-α mRNA with respect to the control (*p* = 0.2185 and 0.2513, respectively), nor the production of NO (*p* = 0.1887 and 0.1953, respectively) nor of TNF-α (*p* = 0.1854 and 0.29881, respectively) in the culture supernatant with respect to the control (Appendix A). 

#### 2.4.3. NeoSTX Inhibits Polarization towards the M1 Phenotype of Murine Macrophages

In order to determine the effect of NeoSTX on macrophage polarization, RAW 264.7 cells were exposed to NeoSTX (1 µM) for 24 h and subsequently cultured with LPS (100 ng/mL) for 18 h. To assess the effect of NeoSTX on M1 polarization, we quantified the expression of the TNF-α mRNA (RT–PCR) and the NO content as an expression of the induction of iNOS by LPS (Griess technique) and the TNF-α protein in the culture of the supernatant (ELISA) and we compared it with the effect obtained by lidocaine (20 µg/mL). As shown in Figure 4A, a significant decrease in the content of TNF-α (*p* = 0.0006 and 0.0002) and NO (*p* = 0.041 and *p* = 0.0325) was observed in the culture supernatant with respect to the LPS situation. In the same way, compared to the previous exposure of NeoSTX or lidocaine to an M1 inducer, a significant decrease in the cellular content of the TNF-α mRNA was observed (*p* = 0.008 and *p* = 0.0012), with respect to the LPS situation.

As a result, we could see that neither NeoSTX nor lidocaine increases the content of NO (*p* = 0.1887 and 0.1422) or TNF-α (*p* = 0.4939 and 0.129) in the culture supernatant, nor does it induce the expression of iNOS mRNA (*p* = 0.5 and *p* = 0.5), IL-1β (*p* = 0.1 and *p* = 0.24), or TNF-α (*p* = 0.5 and *p* = 0.34) with respect to the control. 

#### 2.4.4. NeoSTX Inhibits Polarization towards the M1 Phenotype in the Primary Culture

To determine the effect of NeoSTX on macrophage polarization, equine PBMC were exposed to NeoSTX (1 µM) for 24 h and subsequently cultured with LPS (100 ng/mL) for 18 h. The effect of NeoSTX on M1 polarization was evaluated by quantifying the NO content as an expression of the induction of iNOS by LPS (Griess technique) and the TNF-α protein in the culture supernatant (ELISA) and we compared it with the effect obtained by lidocaine (20 µg/mL). Similarly, we quantified the mRNA expression of iNOS, IL-1β, and TNF-α (RT–PCR). As shown in Figure 4B, in cells pre-treated with NeoSTX or lidocaine and subsequently exposed to an M1 inducer, there were significant decrease in NO content (*p* = 0.015 and *p* = 0.033) and TNF-α (*p* = 0.0001 and *p* = 0.0001) in the culture supernatant and cell content of the iNOS mRNA (p = 0.0051 and *p* = 0.0011), IL-1β (*p* = 0.0048 and *p* = 0.0498), and TNF-α (p = 0.032 and *p* = 0.022), regarding the control situation (LPS; Figure 4C). 

In addition, we evaluated whether, under the same conditions, NeoSTX and lidocaine had the ability to induce M2 polarization. To do this, we evaluated the expression of the equine ARG1 and equine IL-10 mRNA in cultured cells in the presence of NeoSTX (1 µM) or lidocaine (20 µg/mL) for 24 h. As a result, we observed that neither NeoSTX nor lidocaine induces the expression of the mRNA of ARG1 (*p* = 0.314 and *p* = 0.2) or of IL-10 (*p* = 0.242 and *p* = 0.342) with respect to the control (Figure 5). 

### 2.5. NeoSTX Deactivates the M1 Macrophage Phenotype Generated in Response to a known M1 Macrophage Inducer

RAW 264.7 cells were cultured with LPS (100 ng/mL) for 18 h and subsequently exposed to NeoSTX (1 µM) for 24 h. Subsequently, we quantified the expression of the TNF-α mRNA (RT–PCR) and the content of TNF-α and NO in the culture supernatant and compared it with the effect obtained by lidocaine (20 µg/mL). Additionally, a control experiment was carried to quantify the TNF-α content in the cell supernatant in cells cultured with LPS (100 ng/mL) for 18 h, in which the medium was then changed to fresh medium and finally the cells were exposed to NeoSTX (1 µM) for 24 h. As shown in Figure 6A, both exposure to NeoSTX and lidocaine significantly decreased the cellular content of the TNF-α mRNA observed in response to the M1 phenotype inducer being studied (LPS) (*p* = 0.0004 and *p* = 0.042). However, neither NeoSTX nor lidocaine modified the TNF-α or NO content in the supernatant of murine macrophage cultures previously with LPS (*p* = 0.4512 and *p* = 0.075). Nevertheless, when changing the medium and exposing the cells to NeoSTX for 24 h, no increase in TNF-α protein was observed in the culture supernatant. (*p* ≤ 0.0001).

NeoSTX Deactivates the Phenotype of M1 Equine Cells Generated in Response to a Known M1 Macrophage Inducer.

Using the same experimental strategy outlined above, we assessed whether the deactivating effect of NeoSTX on the M1 phenotype was reproducible in macrophages of another species. Accordingly, equine cells were cultured with LPS (100 ng/mL) for 18 h and subsequently exposed to NeoSTX (1 µM) for 24 h. The effect of NeoSTX on the deactivation of M1 was evaluated by quantifying the mRNA expression of iNOS, IL-1β, and TNF-α (RT–PCR) and the content of TNF-α and NO in the culture supernatant and then compared to the effect obtained by lidocaine (20 µg/mL). As shown in Figure 6B, neither NeoSTX nor lidocaine modified the TNF-α and NO content in the culture supernatant (*p* = 0.3301 and *p* = 0.142) in cells pretreated with LPS, but when changing the medium and exposing the cells to NeoSTX for 24 h, no increase in the TNF-α protein was observed in the culture supernatant (*p* ≤ 0.0001).

However, exposure to both NeoSTX and lidocaine significantly decreased the mRNA cell content of macrophage M1 polarization marker genes in response to LPS culture: iNOS (*p* = 0.0236 and *p* = 0.0009), IL-1β (*p* = 0.0143 and *p* = 0.0257) and TNF-α (*p* = 0.0448 and *p* = 0.0095) (Figure 6C).

## 3. Discussion

In this study, the effect of NeoSTX, a specific blocker of voltage-gated sodium channels, in the expression of inflammatory cytokines derived from macrophages treated with LPS, a known polarization inducer towards the M1 phenotype, was evaluated.

The voltage-dependent sodium channels are primarily associated with the support of electrogenesis in excitable cells. However, several investigations have confirmed the presence of functional Nav in cells considered electrically non-excitable but with a substantially lower channel density (<1 versus 2–75 µm², respectively) and whose expression can change markedly depending on the state of development or physiology of the cell [20,24,42,43].

As a first aim, we determined the presence of the different isoforms of the Nav channels (1.1 to 1.9) in a primary culture of equine PBMC. Our results demonstrated that only subtypes Nav 1.5 (tetrodotoxin (TTX-R) resistant) and Nav 1.6 (tetrodotoxin (TTX-S) sensitive) [20,22] are expressed in macrophages derived from equine PBMC (Figure 2A,B), according to what has been observed by different authors in various cell types such as THP-1, macrophages derived from human PBMC, in murine peritoneal macrophages, and in BV-2, microglia and murine macrophages [26,42,43,44].

Likewise, we wanted to determine the location of Nav 1.6 in macrophages of the RAW 264.7 cell line using a rabbit anti-mouse anti Nav 1.6 antibody demonstrating for the first time the immunolocation of the Nav 1.6 isoform only in the intracellular space of these cells where it is widely distributed and not in the plasma membrane as expressed in excitable cells (Figure 2C). This coincides with Craner et al. [45], who observed that the Nav 1.6 isoform is the predominant voltage-dependent sodium channel in both microglia and activated macrophages and with Carrither et al. [43], who described its co-localization with cytoplasmic vesicles and with F-actin polymers in the cytoskeleton. According to the authors mentioned, the role of this isoform is the regulation of cell invasion via modulation of podosome formation, mediating adhesion, invasion, and migration in addition to the polarization of macrophages to M1 and phagocytosis. From the point of view of cell morphology, we could observe in our cells that exposure to NeoSTX evidenced macroscopic changes with respect to cells stimulated with LPS (star shape). Thus, the cells pretreated with NeoSTX tended to maintain the shape of the M0 cells (rounded), and those cells cultured with LPS and those exposed to NeoSTX showed a decrease in their prolongations. However, more specific studies are necessary to demonstrate these changes.

In addition, we have been able to account for the increase in messenger expression of the Nav 1.6 isoform but not of Nav 1.5 in the primary culture when the macrophages being studied have been exposed to LPS (Figure 2E). According to Carrithers et al. (2009) [42], the LPS signal generates changes in the activity of the channels because both the signaling pathways of inflammation, as well as the post-translational modifications and the onset of phagocytosis, generate a depolarizing stimulus sufficient to activate intracellular channels [25,46]. This effect coincides with that of the agonist Veratridine, which generates the opening of the canal [42]. However, the mechanism by which this increase in the expression of Nav 1.6 occurs has not yet been explained. On the other hand, Black et al. (2009) described the co-location of Nav 1.5 in the late endosome, and its role would be associated with the mobilization of positive charges from the endosome to the cytoplasm. These results were corroborated by the authors, using additional channel-specific blockers such as TTX and shRNA channel knockdown [47]. In this regard, it is known that the pharmacological agents acting on the Nav have their receptor in the α subunit and that the receptor site 1 in the pore loop located on the extracellular side of the channel selectivity filter, has specific amino acid residues for the union of guanidine groups of specific blocking neurotoxins that have been used to establish the presence and functionality of Nav in different microglia studies [48]. 

Later, we evaluated whether NeoSTX was able to generate changes in the expression of cytokines of the M1 phenotype. To do this, we first determined whether, in the cells of our study, NeoSTX could be incorporated into the intracellular compartment to bind to the sodium channels located in the cell cytoplasm previously described.

In cells exposed to NeoSTX and subsequently immunostained, we could see that NeoSTX does not bind to any target located in the plasma membrane and that it would bind to targets found in the cell cytoplasm. In studies in which TTX was used, it was suggested that these toxins exert their action by binding to voltage-dependent sodium channels located in the cytoplasm, and that admission to the cytoplasm would occur through pinocytosis or endocytosis/phagocytosis and from this point it would generate the inhibitory effect on the channels [42]. However, this theory has not yet been confirmed.

Another study describes the binding and eventual blockade by TTX and STX to the Cav3 isoform of the type T calcium channel located in the plasma membrane. The basis for this argument would be mainly given by the structural characteristics of the subunits that form the electronegative pore of ionic selectivity, which have a common ancestral origin for both channels, as has been described [49,50]. Thus, guanidine rings from both NeoSTX and TTX would potentially interact with the residues of the selection filter. However, this report has not been confirmed [51], and as we have observed in our study, NeoSTX does not bind to any target of the plasma membrane of the cells being studied.

We also observed that NeoSTX does not modify cell viability in RAW 264.7 cells or in cells derived from equine PBMC (90%; Figure 1), regardless of the concentration of NeoSTX and the exposure time used in this study. In addition, it does not induce any modification by itself in the content of NO or TNF-α in the culture supernatant, nor in the expression of the mRNA of the inflammatory cytokines of the M1 profile iNOS, IL-1β, and TNF-α, nor in the cytokines of the M2 phenotype IL-10 and ARG1 (Figure 5).

Then, to evaluate the effect of NeoSTX on the expression of inflammatory cytokines against an M1 inducer, we analyzed the canonical cytokine mRNA of polarization M1: iNOS, IL-1β, and TNF-α in equine macrophages and TNF-α in RAW 264.7.

The results obtained in our study show that the increase in mRNA production of iNOS, IL-1β, and TNF-α induced by LPS in macrophages of both species is significantly inhibited by pre-treatment with NeoSTX (Figure 4). A similar effect is observed when cells are cultured with LPS and subsequently exposed to NeoSTX in the expression of the inflammatory cytokine mRNA, deactivating the M1 phenotype (Figure 6). The decrease in mRNA expression of these cytokines is consistent with the results obtained by Yuan et al. [40], Lee et al. [21], and Huang et al. [41], who blocked specific VGSC using lidocaine and TTX. In these studies, a decrease in phagocytosis, migration, and expression of pro-inflammatory cytokines such as IL-1α, IL-1β, and TNF-α and also prostaglandin E2 [24,40,44] was observed. They concluded that the mechanisms involved in this lower expression of the mRNAs of the cytokine markers of the M1 phenotype are associated with the inhibition in the activation of TLR-4, the inhibition of the p38MAPK pathway preventing its phosphorylation and the decrease in the activation of NF-kB by inhibiting phosphorylation and the degradation of IKB-α thus avoiding its translocation to the nucleus [21,40]. In addition, from our study, the deactivation effect of TNF-α and iNOS mRNA observed when cells are cultured with LPS and subsequently exposed to lidocaine is greater than that observed for NeoSTX (Figure 4C). This could be due to the lipid-soluble characteristics of lidocaine that facilitate its entry through the plasma membrane to join the receptor site 6 in the inner pore of the VGSC or fenestrations in the lateral wall of the canal [50].

TNF-α protein production and iNOS activity were inhibited when cells were first exposed to NeoSTX and then to the M1 phenotype inducer (LPS), concordant with decreased expression of iNOS mRNA (Figure 4) and TNF-α mRNA, demonstrating that NeoSTX inhibits LPS-induced polarization, noting that the biological action of NeoSTX most likely takes place upstream of the transcriptional activation of LPS-mediated iNOS [52]. Our data demonstrate that NeoSTX is capable of regulating the overproduction of NO in a manner similar to that observed with TTX by Huang et al. [41].

On the other hand, the exposure to NeoSTX, 18 h after stimulation with LPS, did not modify the content of TNF-α or NO in the supernatant. However, when we changed the culture medium before exposure to NeoSTX, we observed a decrease in the content of NO and TNF-α in the supernatant (Figure 6A,B). As described by Venable et al. (2015) [53], TNF-α mRNA expression begins 1 h after LPS stimulation, and protein translation begins after 6 h of exposure to LPS. This explains why we have protein release in the supernatant 18 h after stimulation with LPS and why the subsequent exposure to NeoSTX, reduces the expression of new mRNA. 

The above results showed that NeoSTX prevents M1 polarization and also deactivates the M1 phenotype induced by LPS. In other studies, it has been proposed that the inhibitory effect on M1 polarization is due to the interaction between VGSCs and the TLR-4/NF-kB pathway [24,28,40]. Jung et al. [54] described that the interaction of LPS/TLR4 in microglia induces the expression of Nav 1.1, 1.2, and 1.6 with an increased influx of Na⁺ currents and activation of NF-kB and that TTX inhibits the activation of sodium channels and the translocation of NF-kB to the nucleus and the activation of p38MAPK [54,55]. Additionally, Hossain et al. [44], in microglia and BV-2 cells demonstrated that the blockade of VGSC inhibits the sodium influx given by LPS and decreases both the production of ROS mediated by NADPH oxidase (NOX2) important for the activation of NF-kB, such as the production of TNF-α [44].

Other authors postulate that in T lymphocytes and peritoneal macrophages, the initial increase in intracellular Na^+^ given by VGSC could activate the Na^+^/Ca⁺² exchanger (NCX) in reverse mode through mechanisms not yet well established, generating the influence of Ca⁺² and increasing intracellular Ca^+2^ that activates the canonical pathway of NF-kB and consequently the expression of pro-inflammatory cytokines [20,56,57].

On the other hand, the formation of podosomes would be given by an intracellular signaling mechanism in which Nav 1.6 induces the release of sodium from the intracellular cationic reserves, followed by the entry of sodium into the mitochondria and the subsequent release of mitochondrial calcium through of the mitochondrial Na^+^/Ca⁺² exchanger [42].

The most likely mechanism by which NeoSTX could enter the cytoplasm of these types of cells is through one of its major functions: pinocytosis. Through this mechanism, the toxin would be introduced by invaginations of the plasma membrane that involve particles of the environment and incorporate it into the cell cytoplasm. Once released into the cytoplasm, they would bind to the Nav 1.6 channels. However, future studies are necessary to clarify this point and the underlying mechanisms involved in the observed changes in inflammation cytokines.

In summary, the results of the present study demonstrate the existence of voltage-dependent sodium channels of the Nav 1.5 and Nav 1.6 isoforms in equine cells derived from PBMC and the expression of Nav 1.6 protein in RAW 264.7 cells. On the other hand, we have found that NeoSTX binds to targets located in the intracellular space and not to targets of the plasma membrane and that NeoSTX decreases the expression of the inflammatory cytokines of the M1 profile in response to the induction of polarization by LPS in addition to deactivating the M1 macrophage phenotype.

### 3.1. Conclusions

We can finally conclude that neosaxitoxin inhibits the expression of inflammatory cytokines derived from macrophages by blocking intracytoplasmic voltage-gated sodium channels.

### 3.2. Possible Application

Given this role of modulating inflammatory activity, NeoSTX shows potential use in a wide variety of pathologies with an inflammatory component such as multiple sclerosis, osteoarthritis, cancer, and muscular and immune system disorders.

## 4. Materials and Methods 

### 4.1. Reagents and Chemicals

Neosaxitoxin (NeoSTX) (Laboratory of Membrane Biochemistry of the University of Chile); E. coli lipopolysaccharides serotype 0111: B4 (LPS) (Sigma-Aldrich Chemie^®^ laboratory, Darmstadt, Germany). Rabbit anti-mouse Nav 1.6 antibody (Abcam, catalog ab65166), AB polyclonal anti NeoSTX (Science and Technology Development Foundation, FUCITED, Chile).

### 4.2. Obtaining Ex-Vivo Macrophages from Equines

#### 4.2.1. Inclusion Criteria

The samples for this study were the following: 4 thoroughbred horses, without distinction of sex, between 3–5 years old, associated with Club Hípico de Santiago S.A., who did not present any type of systemic pathology in the two months prior to selection, and without acute musculoskeletal alterations, not undergoing any systemic anti-inflammatory treatments and who are performing their training protocol in a normal way. To verify that there were no subclinical pathologies, a biochemical profile and control hemogram were performed. Any sample that did not meet the previously established requirements were excluded from the study.

#### 4.2.2. Blood Samples

From each of the selected equine specimens, 30 mL of venous blood was collected by trained personnel in EDTA tubes obtained by jugular vein puncture using sterile technique. Blood collection was performed when the samples went to usual blood test procedures (blood count and biochemical profile). From the collected blood, PBMCs were obtained using the Ficoll method according to protocol. (Approval of the ethics committee CBA, Faculty of Medicine U. of Chile, CBA 0832).

### 4.3. Cell Cultures

RAW 264.7 murine cell line cells were seeded at a concentration of 2 × 10⁶ cell/mL and 2 × 10^4^ cell/mL in 6- or 96-well plates in DMEM medium supplemented with 10% FBS, 2 mM L-glutamine, 50 U/mL penicillin G and 50 µg/mL of streptomycin; at 37 °C in a humidified incubator supplemented with 5% CO_2_.

Macrophage derived from equine monocytes were seeded at a concentration of 2 × 10⁶ cell/mL and 2 × 10^4^ cell/mL in 6- or 96-well plates in RPMI-1640 medium supplemented with 10% FBS, 2 mM L-glutamine, 50 U/mL penicillin G and 50 µg/mL of streptomycin at 37 °C in a humidified incubator and supplemented with 5% CO₂. To analyze the purity of the culture, cells derived from PBMC were exposed to plastic for 2 h. They were then washed with PBS, removing the non-adherent fraction [58].

Each culture was exposed to LPS for 18 h and/or NeoSTX for 24 h, before or after treatment with LPS, depending on the used protocols.

### 4.4. Nav Analysis

#### 4.4.1. Primary Tissue Sample

For PCR analysis, brain, heart, striated muscle, dorsal root ganglion (DRG), and liver samples obtained from post-mortem equine destined for human consumption were obtained (Lo Blanco slaughterhouse, Santiago, Chile) and used as positive and negative controls according to the Nav isoform. Tissue samples were cut into 0.5 cm^2^ fragments and placed in 1500 µL Eeppendorf tubes with 500 µL of RNAlater. Subsequently, these samples were transferred to the Physiology Laboratory of the University of Chile, where they were stored at 4 °C for 24 h.

#### 4.4.2. RNA Extraction and Obtaining cDNA for RAW Cells and Equine Cells

RNA extraction was performed using the Trizol reagent method (Total RNA Isolation Reagent, Life Technologies, Carlsbad, CA, USA). Cells were centrifuged at 800 rpm for 10 min; the supernatant was removed and then resuspended in phosphate-buffered saline (PBS). Subsequently, they were centrifuged again at 7500 rpm for 25 s at 4 °C, the supernatant was removed again and 250 µL of PBS, 750 µL of Trizol, and 200 µL of chloroform were added to continue with centrifugation at 12,000 rpm for 10 min. From the result of this process, the aqueous phase was collected and then added to 500 µL of isopropanol and centrifuged again at 12,000 rpm for 15 min at 4 °C. The supernatant was removed, and 75% ethanol was added. After that, the sample was centrifuged at 7500 rpm for 5 min at 4 °C, removing the supernatant to finally resuspend in 10 µL of RNase free water (Invitrogen, Carlsbad, CA, USA). The sample was stored at −20 °C for further analysis.

The mRNA thus obtained was quantified by spectrophotometry, and subsequently, the samples were treated with a DNase kit (Promega, Fitchburg, WI, USA) following the manufacturer’s instructions to eliminate the possibility of genomic DNA contamination.

For the reverse transcription, 1 µL of random primers was added, and then the RNA was treated in a thermal cycler for a cycle consisting of 5 min at 70 °C and another consisting of 5 min at 4 °C. Subsequently, a mix with 5× buffer, 25 mM MgCl2, 10 mM dNTP, RNase, reverse transcriptase, and RNase free water was prepared. Then, 15 µL of this mix was added to each sample. A spin was performed and then treated for 5 min at 25 °C, 60 min at 42 °C, 15 min at 70 °C, and 15 min at 4 °C. The final cDNA product was stored at −20 °C for later analysis.

#### 4.4.3. RT-PCR

The cDNA was amplified using Fast SYBRᵀᴹ Green Master Mix (Applied Biosystems, Foster City, California, USA ), and Nav-specific primers were designed from the NCBI genetic database, in conjunction with the AmplifX 1.5^®^ and PRIMER-BLAST tool (Appendix A). 

The results were normalized with the expression of the β-actin gene, the housekeeping gene. The quantification of the relative change times of the genes being studied was calculated using the 2^△△ Ct^ method and compared with the control group [59].

### 4.5. Cell Viability Tests

Briefly, 2 × 10^4^ RAW 264.7 cells were cultured in Dulbecco’s modified Eagle’s medium (DMEM), at 37 °C with 5% CO_2_ in 96-well plate for 24 h. Subsequently, the medium was changed and NeoSTX was added at 0.1, 1, 10, and 100 nM, and 1 and 10 µM and incubated for 24, 48, and 72 h. Cell viability was assessed using the AlamarBlue technique [40] according to the manufacturer’s instructions. The absorbance was read at 570 and 600 nm using a microplate spectrophotometer (Epoch ™, BioTek, software 2.03., Winooski, VT, USA).

### 4.6. Nitric Oxide Test (NO)

RAW 264.7 cells and cells derived from equine PBMC pretreated with LPS and/or NeoSTX supernatants were collected and the NO concentration was determined using the Griess reagent (Sigma Aldrich, St. Louis, MO, USA). Briefly, the supernatant was collected from the cells and then mixed with the same volume of Griess reagents. The samples were incubated at room temperature for 10 min, and, subsequently, the absorbance was read at 540 nm using a microplate reader [60].

### 4.7. ELISA Test

The concentration of TNF-α in the culture supernatant was determined using DuoSet^®^ Mouse TNF-α (R&D Systems, Minneapolis, MN, USA), and DuoSet^®^ Equine TNF-α (R&D Systems, Minneapolis, MN, USA), according to the manufacturer’s instructions. Subsequently, the absorbance was read at 450 and 540 nm using a microplate spectrophotometer (Epoch™, BioTek, software 2.03.1, Vermont, USA). 

### 4.8. Development of Rabbit Polyclonal Antibodies against NeoSTX

The antiserum against NeoSTX was prepared according to the general procedure described by Torres et al. [61]. Briefly, NeoSTX was coupled to the hemocyanin of *Concholepas concholepas* (CCH; Blue Carrier^®^, Biosonda Corp., Santiago, Chile). In summary, 5 mg of transport protein was dissolved in 2 mL of 0.1 M borate buffer pH 10 and 5 mg of toxin was slowly added in double-distilled water followed by glutaraldehyde at a final concentration of 0.3%. The mixture was incubated for 2 h in the dark at 25 °C, and then dialyzed against phosphate-buffered saline (PBS, pH 7.2) at 4 °C. White female rabbits from New Zealand were immunized by intradermal injection/subcutaneous NeoSTX-CCH emulsified in complete and incomplete Freund’s adjuvant. The presence of specific antibodies against NeoSTX was performed by an indirect enzyme-linked immunosorbent assay (ELISA) with NeoSTX coupled to bovine serum albumin (BSA).

### 4.9. Flow Cytometry

In order to determine the presence of Nav or NeoSTX in both RAW 264.7 cells and in equine PBMC, the cells were cultured with or without NeoSTX (1 µM) for 24 h in DMEM or supplemented Roswell Park Memorial Institute (RPMI) media. Cells were permeabilized with 50 µL of Cytofix/Citoperm (BD, Biosciences, 554722) for 20 min at 4 °C, then washed with 1 mL of Perm/Buffer 1× (Biosciences, 554723) and centrifuged at 700× *g* for 7 min at 4 °C. The pellet obtained was resuspended and subsequently incubated with primary rabbit anti-mouse Nav 1.6 1:100 antibody (Abcam, catalog ab65166) or anti-NeoSTX antibody diluted in Perm/Wash Buffer 1X (Membrane Biochemistry Laboratory, University of Chile) for 10 min. They were then washed with PBS and incubated with a 1:100 FITC anti-rabbit polyclonal secondary antibody (Dakocymation F0205). The samples were analyzed in a Becton Dickinson Cytometer, Canto II model, Diva software (Immunology Laboratory of Faculty of Sciences of the University of Chile).

### 4.10. Confocal Microscopy

Cells being studied were cultured in 12-mm coverslips with or without NeoSTX (1 µM) for 24 h in DMEM or supplemented RPMI media. They were then washed with 1X PBS and fixed with absolute ethanol for 10 min at room temperature. After that, they were washed with PBS 1X and fixed with Cytofix/Citoperm (BD, Biosciences, 554722) for 20 min at 4 °C. They were, subsequently, washed again with Perm/Wash Buffer 1X (Biosciences, 554723) and the primary anti-mouse anti-Nav 1.6 1:100 antibody (Abcam, catalog ab65166) or the anti-NeoSTX antibody (Membrane Biochemistry Laboratory, Universidad de Chile) diluted in Perm/Wash Buffer 1X for 20 min at 4 °C. After this time, the secondary polyclonal anti-rabbit FITC 1:100 anti-body (Dakocymation F0205) was added, washed again with Perm/Wash Buffer 1X and mounted with DAPI 1:200 (BD Pharmingen, 564907) for 15 min. The analysis was performed in a Zeiss brand confocal microscope, model 710, and Zen software was used for data analysis.

## Figures and Tables

**Figure 1 marinedrugs-18-00283-f001:**
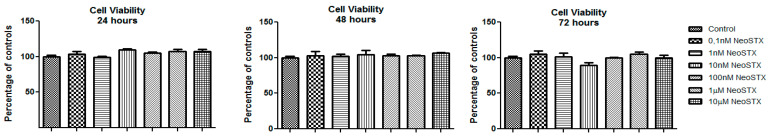
Cell viability. RAW 264.7 cells were treated with Neosaxitoxin (NeoSTX) at concentrations of 0.1, 1, 10, and 100 nM, and 1 and 10 µM for 24, 48, and 72 h.

**Figure 2 marinedrugs-18-00283-f002:**
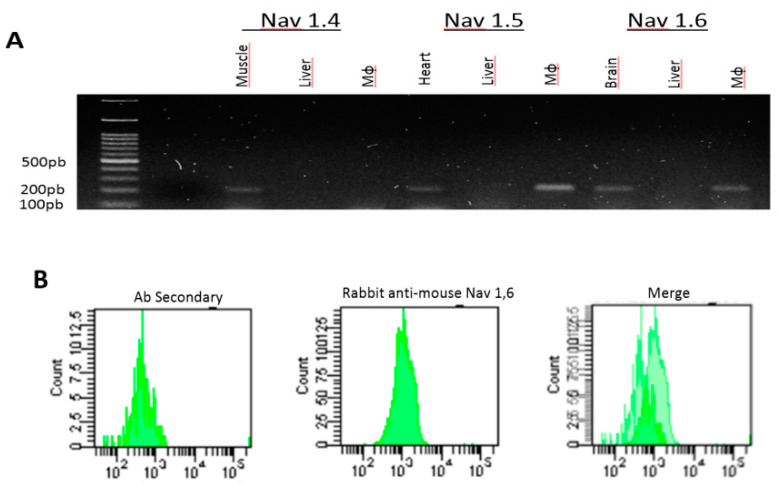
Characterization of the Nav isoforms in a macrophage model. (**A**) Expression of Navs in primary culture. RT–PCR product for isoforms of Nav 1.4, 1.5, and 1.6 in equine PBMC (Mϕ). Positive controls (striated muscle, heart, and brain) and negative control (liver) are shown. (**B**) Representative histogram of flow cytometry in equine PBMC for anti-mouse anti-Nav 1.6 antibody. (**C**) Expression of Nav 1.6 in murine macrophages. Representative histogram of flow cytometry in RAW 264.7 cells for rabbit anti-mouse Nav 1.6. (**D**) Confocal microscopy in RAW 264.7 cells for FITC isotype anti-rabbit antibody, DAPI (blue). Confocal microscopy in RAW 264.7 cells for anti-Nav 1.6 antibody (green), DAPI (blue). (**E**) Effect of NeoSTX on the expression of the equine Nav 1.5 and equine Nav 1.6 mRNA. Equine PBMC were cultured with LPS (100 ng/mL) for 18 h. The expression of the equine Nav 1.6 mRNA was quantified by RT–PCR. β-Actin was used as a control gene. * *p* ≤ 0.05 (*n* = 4).

**Figure 3 marinedrugs-18-00283-f003:**
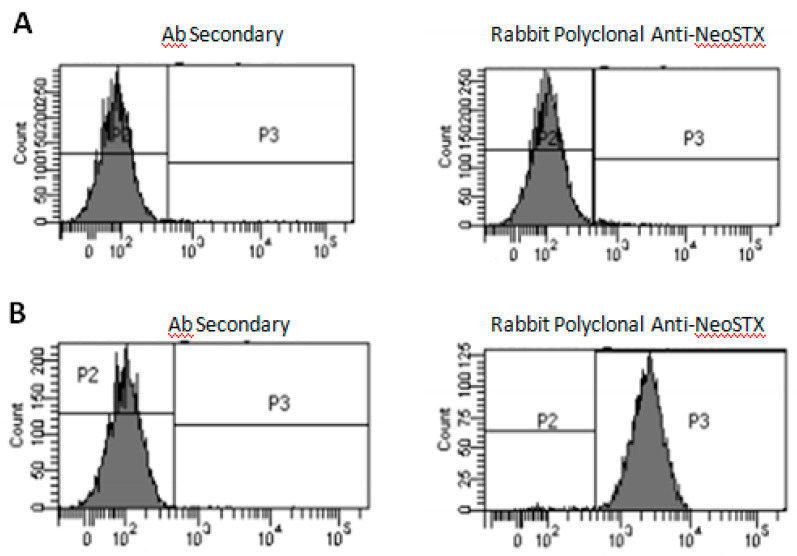
Representative histogram of flow cytometry in RAW cells 264.7 for rabbit anti-NeoSTX antibody. The cells were cultured with NeoSTX (1 µM) for 24 h. (**A**) Non-permeabilized cells, (**B**) permeabilized cells.

**Figure 4 marinedrugs-18-00283-f004:**
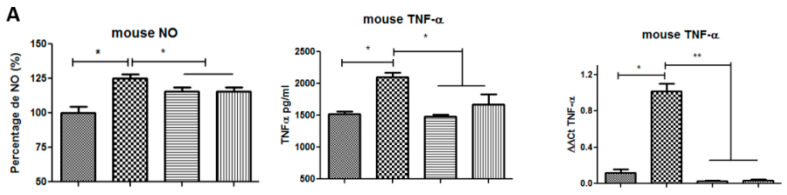
NeoSTX inhibits the expression of markers of inflammation in macrophages. (**A**) RAW 264.7 cells were pre-treated with NeoSTX (1 µM) or lidocaine (Lido) (20 µg/mL) for 24 h and subsequently exposed to LPS (100 ng/mL) for 18 h. The production of NO and TNF-α in the culture supernatant was quantified using the Griess and ELISA technique, respectively, and the expression of mouse TNF-α mRNA by RT–PCR. (**B**) Equine PBMC cells were pretreated with NeoSTX (1 µM) or lidocaine (20 µg/mL) for 24 h and subsequently exposed to LPS (100 ng/mL) for 18 h. The production of equine NO was quantified using the Griess technique and equine TNF-α using the ELISA technique in the culture supernatant. (**C**) The expression of the equine TNF-α, equine iNOS, and equine IL-1β mRNA was quantified by RT–PCR (C). β-Actin was used as a control gene. * *p* < 0.05 ; ** *p* < 0.001 (n = 4).

**Figure 5 marinedrugs-18-00283-f005:**
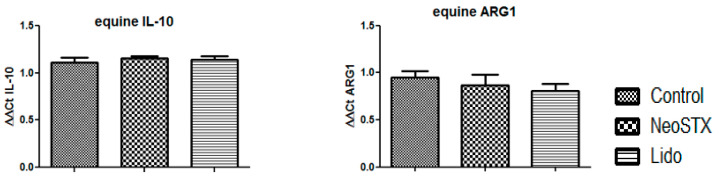
Effect of NeoSTX on M2 phenotype. RAW 264.7 cells were cultured with NeoSTX (1 µM) or lidocaine (20 µg/mL) for 24 h. The expression of the equine ARG1 and equine IL-10 mRNA was quantified by RT–PCR. β-Actin was used as a control gene.

**Figure 6 marinedrugs-18-00283-f006:**
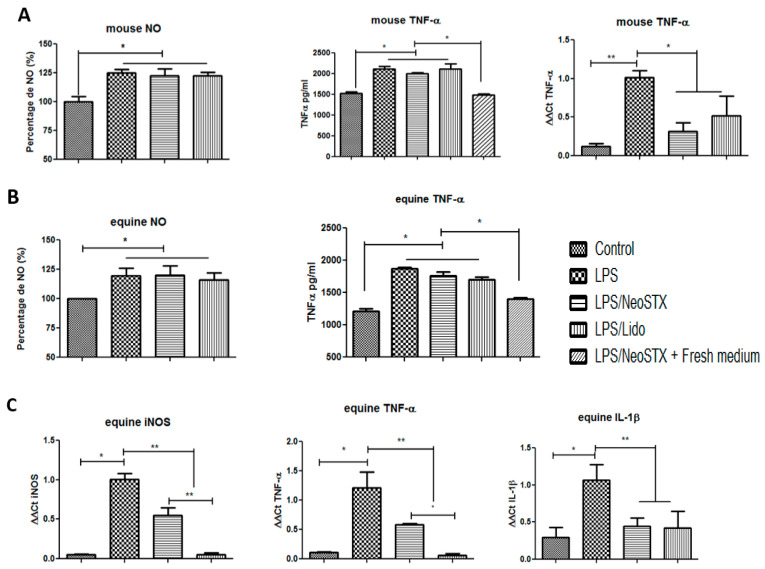
NeoSTX reverses the expression and production of cytokines of the M1 phenotype. (**A**) RAW 264.7 cells were cultured with LPS (100 ng/mL) for 18 h and subsequently exposed to NeoSTX (1 µM) or lidocaine (20 µg/mL) for 24 h. The production of mouse NO and mouse TNF-α was quantified using the Griess and ELISA techniques, respectively, and the expression of the TNF-α mouse mRNA by RT-PCR (A). (**B**) Equine cells were cultured with LPS (100 ng/mL) for 18 h and subsequently exposed to NeoSTX (1 µM) or lidocaine (20 µg/mL) for 24 h. The production of equine NO and equine TNF-α was quantified using the Griess and ELISA techniques, respectively. (**C**) Equine iNOS, equine TNF-α and equine IL-1β mRNA expression were determined by RT–PCR. β-Actin was used as a control gene. * *p* < 0.05, ** *p* < 0.001 (*n* = 4).

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
