# Peer review of "Neosaxitoxin Inhibits the Expression of Inflammation Markers of the M1 Phenotype in Macrophages"

_marinedrugs, 2020, doi:10.3390/md18060283_

Round 1
Reviewer 1 Report
This is a previously reviewed study, submitted as a new manuscript. The authors studied the effect of Neosaxitoxin (NeoSTX), a voltage gated sodium channel blocker, on the production of pro-inflammatory cytokines by M1 phenotype (LPS-stimulated) macrophages, using a murine/macrophage cell line (RAW 264.7) and equine PBMCs. They report that NeoSTX inhibits polarization of M1 macrophages, inhibiting the production of signature cytokines and products, including TNFα, IL-1β and NO. In this version, the authors have addressed critiques to the previous version and include experiments looking at the pre-exposure to LPS, followed by NeoSTX, reporting that the drug was unable to modify the effect of LPS, suggesting that it acts upstream of LPS. The manuscript has also been revised for proper use of English and appears well written. Other controls from the previous version have also been added (e.g., viability studies over a range of concentrations and M2 markers). In general, this is an improved manuscript.
There is, however, an important concern with Figure 6, which doesn’t seem to match the Results described. In fact, Figure 6 looks exactly like Figure 4! Was the wrong figure used?? This needs to be corrected.
There are a few minor critiques:
- The legend of Figure 2 refers to equine PBMC (MÆ ). PBMCs area mix of monocytes and lymphocytes, not only monocytes/macrophages.
- In Figures 4 and 6 (which is the same as Fig. 4), the y-axes are in Spanish (“porcentaje”).
- On line 238, it should be cultured instead of cultures.
Author Response
Response to Reviewer 1 Comments
To facilitate the reading of the modifications, the lines of the corrections are mentioned according to the PDF format. Please see attached PDF.
Point 1: Figure 6 , which doesn´t seem to match the Result described. In fact, Figure 6 looks exactly like Figure 4! Was the wrong figure used?? This needs to be corrected.
R.- We have verified this mistake with editors and it is an editorial fault.
We have corrected the figure 6. See PDF version Please
Point 2: The legend of Figure 2 refers to equine PBMC. PBMCs area mix of monocytes and lymphocytes, not only monocytes/macrophages.
R.- Following the reviewers indication in the first correction, the term "equine cells derived from PBMC" was changed to "equine PBMC". In Material and method Line 471 it is indicated that: To analyze the purity of the culture, cells derived from PBMC were exposed to plastic for 2 hours. They were then washed with PBS removing the non-adherent fraction [58]. The fraction adherent corresponding to macrophages derived from monocytes.
Point 3: In Figure 4 and 6, the y-axes are in Spanich (porcentaje)
R.- The reviewer is correct in his indication and we changed "porcentaje" to percentage in both figures.
Point 4: On line 238, it should be cultured instead cultures
R.- This change is in L 260. cultures → cultured

Reviewer 2 Report
There were still too many typos, gramatical and technical issues to be addressed such as follows:
L42: PAMP recognition receptors (PRR) => pattern recognition receptors (PRRs)
L51: inter alia => among other things, or particularly might be better.
L65: identified. => Please delete the period after identified.
L75: their receptor => their receptors, or their receptor sites
L81: Chile), => Please delete the comma after Chile).
L83: These same => The same, or These similar might be better.
L90: NeoSTX, => Please delete the comma after NeoSTX.
L135, 139 and after: Nav1.6 => Please insert a space between Nav and 1.6.
L184: TNFα => TNF-α
L203: there is a significant decrease in NO content (p = 0.015 and p = 0.033) and TNF-α (p = 0.0001 and p = 0.0001) in the culture supernatant and cell content => there were significant decreases in NO (p = 0.015 and p = 0.033) and TNF-α (p = 0.0001 and p = 0.0001) in the culture supernatant and cell content
L214: equineNO => Please insert a space between equine and NO.
L232: TNF => TNF-α
L247: of: iNOS => Do you need ":"?
L264: expression by => expression was determined by
L283, 296 and after: Nav1.6 => Please insert a space between Nav and 1.6
L304, 400: Nav1.5 => Please insert a space between Nav and 1.5
L365: proinflammatory => pro-inflammatory
L373, 388, 392: No need to return by only one sentence. Please make a paragraph for better flow of the discussion.
L377: microglia, induces => Please delete the comma after microglia.
L378: and, => , and
L376 and throughout: NFkB => NF-kB
L408: Relevance => Possible application?
L630: J Surg Res => No need for the underline.
L633: Na! Channel => Do you need "!"?
All authors shoud check the manuscript throughout to reduce careless mistakes, at least, and to reduce reviewers' burden. Also, the reviewer strongly recommend the authors once again to ask a native speaker of English "with enough scientific writing experiences" to edit this manuscript.
Author Response
Response to Reviewer 2 Comments
To facilitate the reading of the modifications, the lines of the corrections are mentioned according to the PDF format. Please see attached PDF.
Point 1: There were still too many typos, gramatical and technical issues to be addressed such of follows:
R.- We sincerely believe that the errors mentioned are only of a tipographical type and that both the writing and the grammar of the writing were substantially improved. Due to the change in the Word format in which the manuscript arrived, the comments made by the reviewer appear on lines other than those mentioned. To facilitate the reading of the modifications, the lines of the corrections are mentioned according to the PDF format. The final version was revised by a native speaker of English.
L42: This change is found in L43: receptors (PRR) → receptors (PRRs)
L51: This change is found in L52: inter alia => among other things, or particularly might be better.
These pathways allow, inter alia→ These pathways allow, among other things
L65: This change is found in L66: Please delete the period after identified. → have been identified [20,22,23].
L75: This change is found in L76: their receptor → their receptors
L81: This change is found in L82: Chile), → Chile)
L83: This change is found in L84: These same → These similar
L90: This change is found in L91: NeoSTX, → NeoSTX
L135 and after: This change is found in L132 and after: Nav1.6 please insert space between Nav and 1.6. Nav1.6→ Nav 1.6
L184: This change is found in L189: TNFα→ TNF-α.
L203: This change is found in L209: In this line we changed: there is a significant decrease in NO content (p = 0.015 and p = 0.033) and TNF-α (p = 0.0001 and p = 0.0001) in the culture supernatant and cell content of the iNOS mRNA (p = 0.0051 and p = 0.0011), IL-1β (p = 0.0048 and p = 0.0498) and TNF-α (p = 0.032 and p = 0.022), regarding the control situation (LPS) (Figure 4C). → there were significant decrease in NO content (p = 0.015 and p = 0.033) and TNF-α (p = 0.0001 and p = 0.0001) in the culture supernatant as well as in the cell content of the iNOS mRNA (p = 0.0051 and p = 0.0011), IL-1β (p = 0.0048 and p = 0.0498) and TNF-α (p = 0.032 and p = 0.022), regarding the control situation (LPS) (Figure 4C).
Line 214: This change is found in L220: Please insert space between equine and NO. equineNO → equine NO
Line 232: This change is found in L246. TNF → TNF-α
Line 247: This change is found in L262. expression of: iNOS → expression of iNOS
Line 264: This change is found in L293 expression by → expression was determined by
Line 304, 400: This change is found in L325, 426. Please insert space between Nav and 1.5
Nav1.→ Nav 1.5
Line 365: : This change is found in L 375. proinflammatory → pro-inflammatory
Line 377: This change is found in L403. microglia, → microglia
Line 378: This change is found in L404. and, → and
Line 376 and after: This change is found in L404, 405, 408, 413: NFkB → NF-kB
Line 453: This change is in L 436. Relevance → Possible application
Line 696: This change is in line 675 J Surg Res → J Surg Res
Line 633: This change is in L678 Na! →Nav
Point 2: L373, 388,392: No needto return by only one sentence. Please make a paragraph for better flow of the discussion
R.- This change is in L386-399
TNF-α protein production and iNOS activity were inhibited when cells were first exposed to NeoSTX and then to the M1 phenotype inducer (LPS), concordant with decreased expression of iNOS mRNA (Figure 4) and TNF-α mRNA, demonstrating that NeoSTX inhibits LPS-induced polarization, noting that the biological action of NeoSTX most likely takes place upstream of the transcriptional activation of LPS-mediated iNOS [52]. Our data demonstrate that NeoSTX is capable of regulating the overproduction of NO in a manner similar to that observed with TTX by Huang et al [41].
On the other hand, the exposure to NeoSTX, 18 hours after stimulation with LPS, did not modify the content of TNF-α or NO in the supernatant. However, when we changed the culture medium before exposure to NeoSTX we observed a decrease in the content of NO and TNF-α in the supernatant (Figure 6A and 6B). As described by Venable et al., 2015 [53], TNF-α mRNA expression begins 1 hour after LPS stimulation, and protein translation begins after 6 hours of exposure to LPS. This explains why we have protein release in the supernatant 18 hours after stimulation with LPS and why the subsequent exposure to NeoSTX, reduces the expression of new mRNA

Round 2
Reviewer 1 Report
The authors have appropriately addressed the concerns. The mistake involving Figures 4 and 6 has been corrected.
This manuscript is a resubmission of an earlier submission. The following is a list of the peer review reports and author responses from that submission.
Round 1
Reviewer 1 Report
In this study, the authors report studies looking at the effect of Neosaxitoxin (NeoSTX), a voltage gated sodium channel blocker, on the production of pro-inflammatory cytokines by M1 phenotype (LPS-stimulated) macrophages, using a murine/macrophage cell line (RAW 264.7) and equine PBMCs. They report that NeoSTX inhibits polarization of M1 macrophages, inhibiting the production of signature cytokines and products, including TNFa, IL-1b and NO.
While the manuscript has good potential, there are several concerns that need to be addressed:
- Although relatively well-written, the manuscript, if accepted for publication, needs to be revised for proper use of English.
- The authors state in their abstract that NeoSTX does not compromise the macrophage polarization to the M2 phenotype. Yet, there is no data presented in this study to support that assertion. They should refrain from doing this.
- The effect of NeoSTX on cell viability is a critical control in this study. Even though the authors claim that it did not significantly affect viability, results over a range of concentrations and time, should be shown.
- Some of the figures are confusing, and the authors should make an effort to label them appropriately. For example, in Figures 1B and 1C, 2A, 2B the figures are labeled by tube numbers, which doesn’t allow the readers to quickly know what the represent. Please label them with the experimental condition they represent.
- Sometimes, the authors refer to the cells as “equine cells derived from PBMC, which is confusing. They should simply refer them as “equine PBMC”. In some cases, the refer them to PBMC (MÆ ). PBMCs area mix of monocytes and lymphocytes, not only monocytes/macrophages.
- The legend to Figure 1A states that several tissues are shown (e.g., brain, striated muscle, heart, dorsal root ganglion). The figure shows only brain, liver and monocytes.
- The authors refer to the antibodies several ways. This should be standardized. The antibody is a rabbit anti-mouse Nav1.6.
- Did the authors use a horse-specific TNFa ELISA? Only the commercial mouse TNFa ELISA is mentioned in materials and methods. We also assume that they are using the rabbit anti-mouse Nav1.6 antibody to stain the equine cells as well. Is this correct? Is there a reference to support cross-species reactivity?
- How do the authors explain the apparent discrepancies between the protein and the mRNA data in Figure 4 (e.g., inhibition at the mRNA but not the protein level?). This may be due to the culture conditions, in which the cells were first cultured with LPS for 18 hours before addition of the NeoSTX. A preferred condition would have been to harvest the cells after LPS stimulation and put them in fresh medium with the NeoSTX.
- Figure 3 and 4 have subtitles “equine”and “equino”. Only English sould be used.
Reviewer 2 Report
The data of this study are interesting, but figures are not clear, and there are too many typos and careless mistakes such as follows to correct. It is recommended to ask a native speaker of English with enough scientific writing experience to edit this manuscript throughout.
L14: Just before the abbreviation "NeoSTX", full spelling "Neosaxitoxin"is needed like "Neosaxitoxin (NeoSTX)".
L18, 19 and throughout: content => concentration in the cell culture supernatant? Please specify.
L36: Please delete an extra space from [ 3] to [3].
L41: A period is needed after [5,6] like "[5,6].".
L54: transcription IL-10 => transcription of IL-10?
L57: include => includes, pro- inflammatory => pro-inflammatory
L59: excessive damage host prevention =>excessive damage, host prevention, or more understandable expression is there?
L66, 69, 75, 102, 136, and throughout the manuscript: The period should be placed at the end of the sentences or after the parentheses.
L84: The reference is needed for Lobo et al, in 2015 in a clinical safety study for NeoSTX.
L111: PBMC equine => equine PBMC
L141: 18h => 18 h or 18 hours as the other parts.
L218, 358: TNFα => TNF-α
L353: and, and => either can be deleted.
L360, 361 and throughout the manuscript: Ca +2 => Ca+2 or Ca2+
L372: Future => future
L414: sample => samples
L454: was => were
References: Please follow to the formats of the Journal.
L522: Inmunol => Immunol
Reviewer 3 Report
The study "Neosaxitoxin inhibits the expression of inflammation markers of the M1 phenotype in macrophages" by Montero et al., tried to prove that NeoSTX abrogates the inflammatory pathway while the evidences they provided were not enough to draw that conclusion.
Authors need to provide additional data and clarify some of their data presented in the manuscript.
- In Fig 1A authors showed expression of Nav1.1 to Nav1.9 channels while authors did not pride primer sequences and nothing mentioned about the amplified sequence size, of how many base pairs were the PCR products. Marker should be labelled as bp. Further authors run samples on different gels an then join the figures with one market which is not the correct way of presentation, each get should be presented with its own marker. Two bands are visible in each amplification why?
- Figure 1: Authors stained the cells for Nav1.6, they should perform the staining again, quality is very poor, cannot be concluded that there is Nav1.6 expressed, repeat the staining and also provide the negative control.
- Authors claimed that Nav1.6 protein expression is increased while no significant change in Nav1.5 mRNA in RAW 264.7 cells while did not show the data. Data must be shown
- Almost in 60% of result sections authors claiming no significant changes or significant change but mentioned DATA NOT SHOWN. Authors cannot claim something when they are not presenting the data. Data must be presented from the experiments.
- Authors mentioned, "To determine if NeoSTX has an effect on the viability on cells exposed to it, 2 * 104 RAW 264.7 cells were treated with NeoSTX at concentrations of 0.1nM, 1nM, 10nM, 100nM, 1μM and 10μM for 24, 48 and 72 hours. The AlamarBlue technique was used to assess cell viability [40]. In all experimental conditions, cell viability was greater than 90% regardless of the concentration of NeoSTX used and the exposure time. (Data not shown)". Again data was not presented, authors must show the data to support their claim.
- Discussion is not supported by the results. Most of the parts of discussion are speculative, no concrete evidence has been presented. Authors did not show investigated the NFkB, MAPK and STAT pathways in their experimental model but linked those in a speculative way to their results and hypothesis.